# A protocol for a pilot randomised controlled trial of an Early Psychiatric Assessment, Referral, and Intervention Study (EPARIS) for intensive care patients

Dylan Flaws [1,2,3,4] *, Chelsea Allen[5], Stuart Baker[6], Adrian Barnett [5], Olivia Metcalf[7], Hamish Pollock[6], Mahesh Ramanan [8,9,10,11], Alexis Tabah[2,4,6], Tracey Varker[7]

**1** Critical Care Research Group, The Prince Charles Hospital, Queensland, Australia, **2** School of Clinical Sciences, Queensland University of Technology, Queensland, Australia, **3** Metro North Mental Health, Caboolture Hospital, Queensland, Australia, **4** Faculty of Medicine, University of Queensland, Queensland, Australia, **5** Australian Centre for Health Services Innovation and Centre for Healthcare Transformation, School of Public Health & Social Work, Queensland University of Technology, Queensland, Australia, **6** Department of Intensive Care, Redcliffe Hospital, Queensland, Australia, **7** Phoenix Australia, Centre for Posttraumatic Mental Health, Department of Psychiatry, University of Melbourne, Victoria, Australia, **8** Department of Intensive Care, Caboolture Hospital, Queensland, Australia, **9** Adult Intensive Care Services, The Prince Charles Hospital, Queensland, Australia, **10** Mayne Academy of Critical Care, University of Queensland, Queensland, Australia, **11** Critical Care Division, The George Institute for Global Health, University of New South Wales, New South Wales, Australia

* Dylan.flaws@health.qld.gov.au

**Data Availability Statement:** No datasets were generated or analysed during the current study. All

## Abstract

### Background

Up to 80% of Intensive Care Unit patients experience physical, cognitive, and/or psychological complications post-discharge, known as 'Post Intensive Care Syndrome' (PICS). Early diagnosis and intervention are a priority, but while current post-intensive care follow-up processes endorse a multidisciplinary model, incorporating a psychiatric consultation has not been studied.

### Methods

A pilot, open-label randomised controlled trial was developed by a multidisciplinary team to evaluate the feasibility and acceptability of incorporating a psychiatric review into an existing post-ICU clinic. The study will run for 12 months and aim to recruit 30 participants. Inclusion criteria for participants: a) ICU admission greater than 48 hours, b) no cognitive impairment that prevents participation, c) ≥ 18 years old, d) residing in Australia, e) fluent in English, f) able to provide GP information, and g) likely to be contactable in 6 months. Patient recruitment will be at Redcliffe Hospital, Queensland, Australia, and will involve patients attending the Redcliffe post intensive care clinic. Participants will be allocated to intervention or control using block randomisation and allocation concealment. Participants allocated to the control arm will receive the standard cares provided by the clinic, which involves an unstructured interview about their ICU experience and a battery of surveys about their psychological,

relevant data from this study will be made available upon study completion.

**Funding:** DF, MR Metro North Hospital and Health District Clinician Research Fellowship https://metronorth.health.qld.gov.au/research/grants/crf The funders had and will not have a role in study design, data collection and analysis, decision to publish, or preparation of the manuscript.

**Competing interests:** The authors have declared that no competing interests exist.

cognitive, and physical function. Those allocated to the intervention arm will receive these same cares as well as an appointment with a psychiatrist for a single session intervention. The psychiatric intervention will involve a comprehensive review, including comorbid disorders, substance use, suicidal ideation, psychosocial stressors, social/emotional supports. Psychoeducation and initial treatment will be provided as indicated and recommendations given to the patient and their GP about how to access ongoing care. In addition to surveys conducted as part of standard clinic cares, all participants will complete additional questionnaires about their history, hospital experience, mental and physical health as well as employment circumstances. All participants will be followed up 6 months after their appointment and will be invited to complete follow-up questionnaires about their mental and physical health, as well as health service use and employment circumstances. The trial has been registered with ANZCTR (ACTRN12622000894796).

## Results

To evaluate the feasibility and acceptability of the intervention to the patient population. Differences between groups will be assessed using an independent samples t-test. Resource requirements to administer the intervention will be evaluated by reporting the mean duration of the EPARIS assessment and approximate cost per patient to provide this service. To estimate the effect size of any treatment effects, changes in secondary outcome measures between baseline and 6 months will be compared between intervention and control groups using Analysis of Covariance regression. As this is a pilot, we will not use p-values or test a null hypothesis, but will give confidence intervals.

## Conclusions

This protocol provides a pragmatic evaluation of the acceptability of introducing early psychiatric assessment into an existing post-ICU follow-up process, and if considered acceptable will inform future research into the efficacy and generalisability of the intervention. The strengths of EPARIS are the prospective, longitudinal design with a control population, and its use of validated post-ICU outcome measures.

## 1. Introduction

Novel technology, and clinical research have consistently improved Intensive Care treatment to the point where around 90% of patients admitted to an Intensive Care Unit (ICU) today will be discharged alive [1]. This rise in survival rates has precipitated a shift in focus from mere survival, to the quality of life after survival. It is now known that post ICU recovery is often complicated and partial [2], with as many as 80% of patients experiencing some form of physical, cognitive, and/or psychological complication which can persist for up to 15 years after their ICU discharge [3–5]. This complicated recovery trajectory following an ICU admission has been described in literature as 'post intensive care syndrome' (PICS).

Mental illness is a common feature of PICS. A large UK cohort study of nearly 5,000 ICU survivors found the prevalence of anxiety, depression, and PTSD to be 46%, 40% and 22% respectively, with 18% meeting criteria for all three [6]. An increased prevalence of mental illness treatment and psychotropic medication use was also recently found in the 5 years following an ICU admission when compared to the 5 years prior [7].

The prevalence and burden of PICS may vary between populations, but it is demonstrably associated with substantial personal and financial burden for patients and families [8]. Disrupting social engagement and employment, the health and social costs are immense [3, 9, 10] and post-ICU psychological morbidity is associated with poor quality of life (QoL) across all domains regardless of aetiology [11].

Further research into prevention, early diagnosis and intervention of PICS is internationally considered a priority [12]. Many post-ICU clinics review patients after discharge, often with nursing, allied health, and psychologist input, but to our knowledge the effect of a consultation with a psychiatrist soon after ICU discharge on PICS outcomes has not been investigated. This is despite a recent meta-analysis demonstrating benefit from early psychological intervention targeted to symptomatic individuals following recent trauma [13] and a recent overview of PICS emphasising early screening and treatment of psychological morbidity to prevent longer term impairment, and advocating for a more targeted approach to intervention delivery [14].

The aim of the current paper is to describe the protocol for the Early Psychiatric Assessment, Referral, and Intervention Study (EPARIS). This randomised controlled trial will compare early psychiatric assessment and referral to treatment as usual. It is hypothesised that an early psychiatric consultation involving evaluation, initial treatment (such as psychoeducation and pharmacotherapy), recommendations on how to access ongoing care will be feasible to implement in a post-ICU clinic and acceptable to that population. Secondary aims include testing whether an intervention may improve a patient's sense of self-efficacy, improve access of ongoing therapy, and reduce the long-term burden of PICS.

## 2. Objectives

The primary objective of this study is to:

1. To evaluate feasibility (cost and practicability) and acceptability of an early psychiatric intervention in a post-ICU clinic.

   Secondary objectives are:

1. Assess the impact of early psychiatric assessment and provision of referral options on self-efficacy and rates of health service access for patients requiring psychological intervention post-ICU.

2. Measure the change in symptom severity for anxiety, depression, and PTSD between presentation, and 6 months after presentation.

3. Measure the impact on work, income, and independence in activities of daily living (ADL).

## 3. Materials and methods

### 3.1 Protocol version

Version 3 dated 06/09/2022

### 3.2 Ethical and research approvals

Ethics approval has been obtained by The Prince Charles Hospital Human Research Ethics Committee (Approval number: HREC/2020/QPCH/86123) and protocol is reported in accordance with SPIRIT-PRO guidelines [15].

The trial has been registered with ANZCTR (Approval number: ACTRN12622000894796)

### 3.3 Study design

The EPARIS study is a prospective, parallel, single site, pilot, two-arm randomised controlled trial. The effectiveness of early psychiatric assessment, referral and intervention will be compared with treatment as usual. Data will be collected with consent from routinely kept hospital records, and from participants at two times (pre-intervention and six months post intervention).

This trial will be conducted and reported according to the pilot trials extension of the CONSORT statement: http://www.consort-statement.org/extensions/overview/pilotandfeasibility.

**3.3.1 Setting.** This study will be conducted in the Redcliffe Hospital, Queensland, Australia. The Redcliffe Hospital ICU has 10 beds and admits 500 patients per year. It provides general intensive care management for a broad spectrum of critical illnesses, apart from tertiary cardiothoracic, neurosurgical, trauma and burns management.

Redcliffe Hospital has an established ICU follow-up clinic, where ICU survivors are reviewed around three months after discharge from hospital and data pertaining to physical, cognitive, and psychological functioning, quality of life and disability is collected to monitor for PICS. Where specific issues are identified, appropriate counselling and referrals are provided. Patients are referred predominantly from the Redcliffe Hospital ICU, but also from other local hospital ICUs.

The clinic reviews approximately 80 patients per year. All ICU discharges meeting eligibility criteria are contacted by telephone between 1 and 3 months after ICU discharge and offered a post-ICU clinic appointment. The clinician makes initial contact with patients recently discharged from ICU who meet engagement criteria (ICU admission over 48 hours and able to provide informed consent).

### 3.4 Study team, participants and sample size

The study team was created through and existing academic partnership between the Psychiatrist (DF) intended to deliver the intervention and ICU staff operating the current Post-ICU Clinic (SB, CA, AT, HP) who will be responsible for recruitment and data collection. Content experts OM and TV were approached to ensure intervention fidelity and best-practice. Methods and process expertise was sought from statistician AB and Intensivist MR.

Participants will be sourced from patients attending the clinic. This study is aiming for a convenience sample of 40 participants. The feasible recruitment window is 12 months, with an estimated maximum possible sample size of 80. Assuming a 50% recruitment percentage (with 70% agreeing to participate and 70% of those completing the follow-up survey), it is estimated that data will be collected from a sample of approximately 30 to 40 patients attending the Redcliffe ICU follow-up clinic over 12 months, with 15 to 20 allocated to the control and intervention arms respectively.

### 3.5 Inclusion and exclusion criteria

People discharged from the participating ICU will be eligible to participate if they meet the inclusion criteria described in Table 1.

### 3.6 Initial approach and consent

Eligible patients will be contacted by clinical staff and invited to attend the clinic 1 to 3 months after their ICU discharge, as summarised in Fig 1. If a patient agrees to attend the clinic, the staff member will inform them about the study. If the patient expresses interest, verbal

**Table 1. EPARIS inclusion and exclusion criteria.**

| Inclusion | Exclusion |
|---|---|
| An ICU admission > 48 hours | Life expectancy at discharge < 6 months |
| No cognitive impairment that prevents participation | Unable or unwilling to provide consent to participate for the study duration |
| ≥ 18 years | Unlikely to remain contactable for study duration |
| Sufficient English | |
| Willing to provide GP information | |

permission will be sought to allocate the patient's appointment time with a view to conducting full consent to participate upon attending their appointment.

Upon attending the clinic, potential participants will be provided written and verbal information regarding the study and invited to ask any questions they have before being invited to participate. Potential participants will not be aware if they have been allocated to the control or intervention arm at this stage, but staff and investigator concealment will not be possible beyond this point due to the psychiatrist being present/absent in the clinic.

A written patient information and consent form will be provided. People who agree to participate will complete the written consent form before baseline data collection. Participants will also give permission for research staff to access data from their clinical records and will complete a series of questionnaires as described above. Eligible, fully informed, and consenting patients will then be entered into the study (see Fig 2).

### 3.7 Allocation and concealment

Patients who provisionally agree to participate will be allocated to attend appointments when either only business-as-usual cares are available, or when the intervention also is available. We will use block randomisation in blocks of 4 or 6, in a 1:1 ratio, with a computer-generated list by the study statistician (AB) and provided in enclosed envelopes to conceal allocation until verbal consent is provided. An appointment time will then be provided, but allocation to treatment or control will remain concealed to the participant at this stage. This will be an open label RCT as will not be possible to conceal whether participants are in the intervention or control arm due to the nature of the intervention.

Patients who choose not to participate will be offered appointment times as per business-as-usual and will not be part of the study. To monitor recruitment, a re-identifiable screening and recruitment log will be kept that will document patients eligible for approach, and subsequent contact. If a participant that has been allocated to receive the early psychiatric intervention chooses to withdraw consent, they will receive the standard cares provided by the clinic without the intervention, and any collected data will be withdrawn from the study.

### 3.8 Data collection

The surveys described in Table 2 below have been selected to provide information on the patient's background (premorbid risk/protective factors and baseline level of function), PICS measures (psychological, physical, and cognitive function) and Post appointment assessments (acceptability of the appointment, and subsequent health service use patterns). The battery of questions conducted at the appointment and at 6 months were both tested with a control sample, finding that completion takes 13 to 20 minutes. Data will only be accessible to members of the research team.

| | Study Period | | | | |
|---|---|---|---|---|---|
| | **Enrolment** | **Allocation** | **Post-allocation** | | **Close-out** |
| **Timepoint** | **Phone call invitation to Clinic** | **Post verbal consent to allocation** | **Pre-appointment** | **Post-Appointment** | **6 months Post Appointment** |
| **Enrolment** | | | | | |
| Eligibility Screen | X | | | | |
| Informed Consent | | X | | | |
| Allocation | | X | | | |
| **Interventions** | | | | | |
| Consent to Continue | | | X | | |
| BAU | | | ← | → | |
| EPARIS | | | ← | → | |
| **Assessments** | | | | | |
| Background Information | | | X | | |
| Demographics | | | X | | |
| Patient Reported Experience Measure (**PREM**) | | | X | | |
| Life Events Checklist (**LEC-5**)[16] | | | X | | |
| Posttraumatic Adjustment Screen (**PAS**) | | | X | | |
| General Self Efficacy Scale[17] | | | X | | X |
| Employment Information Questionnaire | | | X | | X |
| **Outcome Variables** | | | | | |
| Hospital Anxiety and Depression Scale (**HADS**)[18] | | | X | | X |
| PTSD Checklist for DSM-5 (**PCL-5**)[19] | | | X | | X |
| **EQ5D-5L**[20] | | | X | | X |
| Montreal Cognitive Assessment (**MOCA-BLIND**)[21] | | | X | | |
| Post Appointment Questionnaire | | | | X | |

**Fig 1. SPIRIT schedule.**

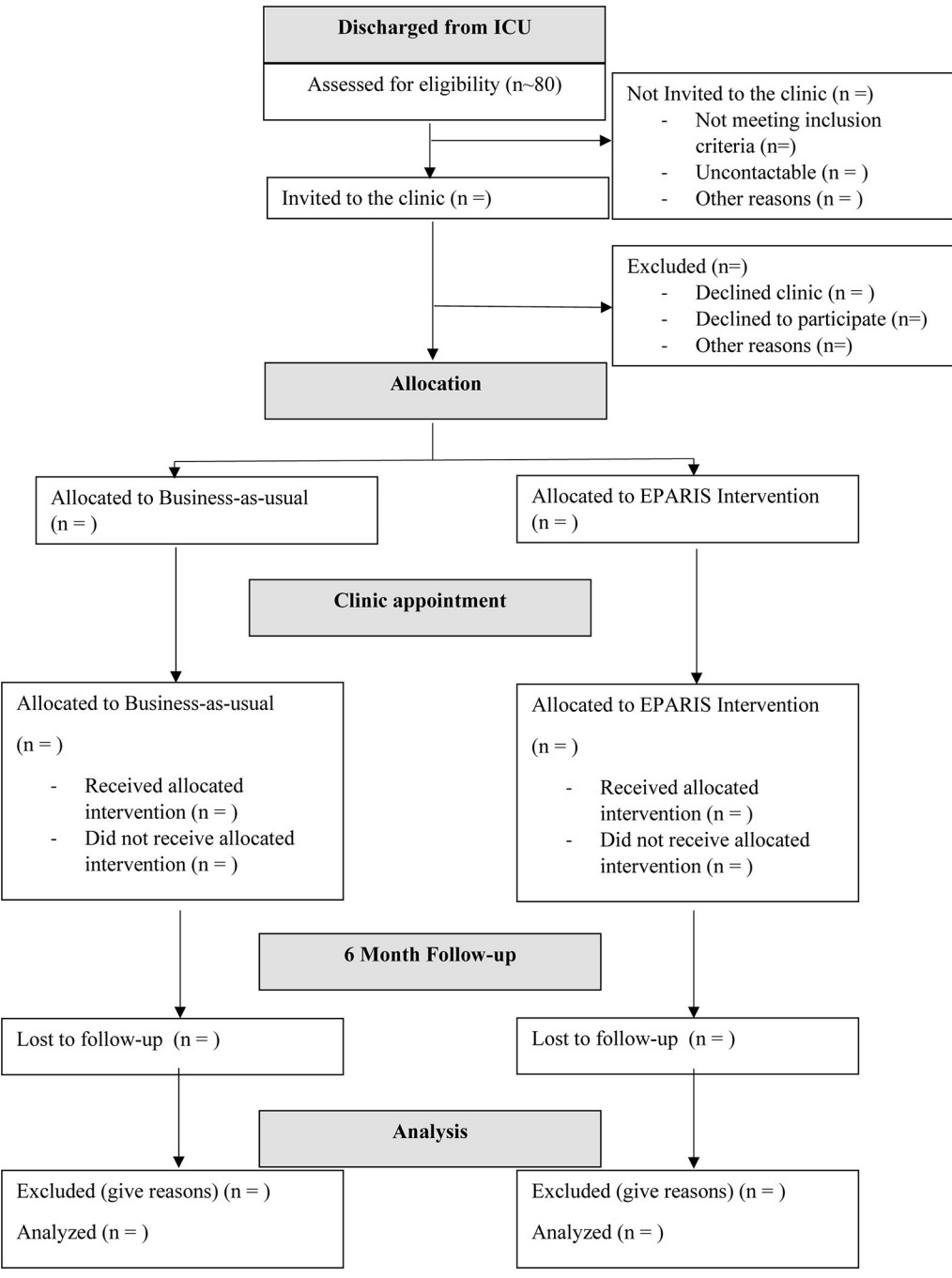

**Fig 2. Recruitment flowchart.**

## 3.9 Clinic interventions

**3.9.1 Treatment as usual.** At their appointment, all patients will complete standardised tools related to physical and mental wellbeing and participate in an interview (See Table 2) followed by the opportunity to provide feedback to the service about their hospital and clinic experience. The standard interview process is unstructured and guided by the patient, but generally covers the topics summarised in Table 3.

**Table 2. Surveys conducted during the study.**

| Survey | Structure | Description | Routinely Collected | Collected for Study | Collected at 6-month follow-up |
|---|---|---|---|---|---|
| **Background Information** | | | | | |
| Patient Reported Experience Measure (**PREM**) | 11 item self-reported survey | Used to collect information about the participant's experience during their hospital admission. | | X | |
| Life Events Checklist (**LEC-5**) [16] | 16 item self-reported survey | Designed to screen for potentially traumatic events in a respondent's lifetime. | | X | X |
| Posttraumatic Adjustment Screen (**PAS**) | 10 item self-reported survey | Designed to identify individuals at risk of developing PTSD/depression following a traumatic injury. | | X | |
| General Self Efficacy Scale [17] | 10 item self-reported Survey | Designed to assess optimistic self-beliefs to cope with a variety of difficult demands in life. | | X | X |
| Employment Information Questionnaire | 7 item self-reported survey | Explores changes to employment and income since their admission. | | X | X |
| **PICS Measures** | | | | | |
| Hospital Anxiety and Depression Scale (**HADS**) [18] | 14 item self-reported survey in 2 subscales | Measures symptoms of anxiety and depression | X | | X |
| PTSD Checklist for DSM-5 (**PCL-5**) [19] | 20 item self-reported survey | Assesses the 20 *DSM-5* symptoms of PTSD. | | X | X |
| **EQ5D-5L** [20] | 5 item self-reported survey | Non-disease specific instrument for measuring health-related quality of life. | X | | X |
| Montreal Cognitive Assessment (**MOCA-BLIND**) [21] | 10 item clinician administered test | Rapid screening instrument for mild cognitive dysfunction. | X | | |
| **Post Appointment Assessments** | | | | | |
| Post Appointment Questionnaire | 7 item self-reported survey | Questions related to the perceived acceptability of their appointment | | X | |
| Health Services Use | 12 item self-reported survey | Explores access and utilisation of various health services in the preceding month. | | | X |

**3.9.2 Early psychiatric assessment and referral (EPARIS) intervention.** EPARIS consists of an assessment phase, personalised treatment recommendations, and referrals to relevant services.

*3.9.2.1 Assessment.* Participants randomised to the intervention group will be given an appointment on a day and time when the psychiatrist is in attendance. Participants will see the psychiatrist in addition to completing the treatment-as-usual process described above.

Prior to assessment, the psychiatrist will review clinical notes pertaining to premorbid history, and their ICU admission. In addition, the psychiatrist will review the baseline

**Table 3. Content of treatment-as-usual clinical interview.**

| History | The electronic medical record and discharge summary will be reviewed. Events surrounding the patient's initial presentation are used to begin the conversation about admission. ICU events such as issues and interventions encountered are discussed. Staff are guided by the patient, and respectful when a patient chooses not to revisit their admission in detail |
|---|---|
| Systems Assessment | A review of body systems (eg. Cardiovascular, musculoskeletal, including mental health) is conducted, then the patient self-assesses their quality of life and functional ability pre and post ICU. Further assessment and referral to specialist services is discussed as appropriate. |
| Psycho-social | Social history encompassing pre and post ICU admission is explored, including relationships, employment, housing, and social interactions. Staff review services that may be able to alleviate identified stressors. The patient's recollection of their admission is explored, mindful that some patient recollections will vary. Staff acknowledge patient feelings and experiences by explaining the prevalence and causes (eg. hallucinations and delusions) and consider referral for counselling services. |

questionnaires, ICU notes and other relevant clinical information. The psychiatrist will discuss these with the participant and explore how this interrelates with their premorbid history, and their ICU experience.

The psychiatrist will conduct a comprehensive psychiatric review, including comorbid disorders, substance use, suicidal ideation, psychosocial stressors, social/emotional supports. Diagnoses of any mental illnesses present will be made based on DSM-V criteria. Psychoeducation will be provided on any new mental illness present, detailing how this interrelates with their critical illness recovery, and general lifestyle advice will be provided. Initial treatment for any new mental illness commonly seen in post-ICU populations will be provided in accordance with local psychiatric guidelines.

*3.9.2.2 Psychological treatments.* Where indicated, education will be provided on appropriate psychological treatments. A letter will be sent to the patient's GP to make referral for the agreed-on therapy.

*3.9.2.3 Pharmacological treatments.* The risks and benefits of pharmacological treatment will be discussed. If commenced, a letter will be sent to the patient's GP to follow-up response and tolerability. Time taken for the intervention will vary between patients and will be documented, but up to one hour will be allowed for each patient.

*3.9.2.4 Mental illness not related to ICU presentation.* Where a patient has a pre-existing mental illness which is already being adequately managed, no further intervention will be provided. If new mental illness is identified or suspected, but unrelated to their ICU admission, contact details for relevant services will be provided and the psychiatrist will notify the patient's GP.

*3.9.2.5 Intervention follow-up.* The Psychiatrist will follow-up all patients where an intervention is initiated 4 to 6 weeks after their appointment. If there has been at least a partial clinical response, a letter will be sent to the participant's GP recommending ongoing monitoring. If there has been no response or a non-urgent deterioration, the letter will suggest referral for ongoing psychiatric review.

*3.9.2.6 Imminent risks.* If the participant appears to require inpatient psychiatric treatment, or there is concern of an imminent risk (such as suicidal ideation with intent and plan), or an urgent clinical deterioration, the participant will be directed to the local Emergency Department for review. If the imminent risk is identified during the intervention follow-up, the participant will be encouraged to attend their local hospital for urgent review.

## 3.10 Follow-up at 6 months

Six months after clinic presentation, research staff will contact all participants to offer a follow-up appointment either in the clinic or by phone and conduct the follow-up survey as described below. If participants do not respond to the first contact attempt, research staff will attempt contact again one week later.

Participants will complete a combination of some repeated measures from baseline, and some new measures as described in Table 2 above.

## 3.11 Fidelity of intervention

The intervention will be administered by a single psychiatrist (DF) who will meet regularly with content experts (OM and TV) to discuss scenarios arising from the study, and how best to maintain intervention fidelity in each scenario. For situations not adequately accounted for in the protocol, a decision will be made by this group and documented as a precedent for future scenarios. This will be published with the results of the study in a deidentified format.

## 4. Statistical plan

Analysis of this study aims to:

1. Evaluate the feasibility and acceptability of the intervention to the patient population

2. Estimate the effect size of any treatment effects to inform power calculations of future studies evaluating whether the intervention is beneficial

### 4.1 Outcome measures

We will not use a primary outcome as this is a pilot study and hence the results are primarily to inform a larger design rather than confirming any hypothesis. This pilot, will also not use p-values or test a null hypothesis, but will provide relevant confidence intervals.

**4.1.1 Feasibility and acceptability outcomes.** Responses to the Post Appointment Questionnaire will be assumed to be additive and averaged to produce an outcome measure of acceptability. Differences between groups will be assessed using an analysis of covariance (ANCOVA) model with the baseline questionnaire as an independent variable. A clinically significant finding in favour of the control arm will indicate a lack of acceptability for the EPARIS intervention, whereas no difference, or a difference in favour of the EPARIS arm will indicate acceptability.

Resource requirements to administer the intervention will be evaluated by reporting the mean duration of the EPARIS assessment and approximate cost per patient to provide this service as a measure of feasibility. Principal costs are likely to be human resources, but other considerations such as space and materials will also be described. It is anticipated that a process that allows 4 to 8 patients to complete the EPARIS intervention within a 4-hour clinic session would constitute a feasible process. As with acceptability, differences in resource requirements will be compared between the control and intervention groups.

**4.1.2 Treatment outcomes.** Symptom scores for Anxiety, Depression, PTSD, and Physical functioning will be completed at the clinic appointment and repeated at 6 months. Anxiety and depression will be measured with the Hospital Anxiety and Depression Scale (HADS), PTSD symptoms with the PTSD Checklist for DSM-5 (PCL-5), and physical function with the EuroQOL survey (5 domains, 5 levels) EQ-5D-5L.

Changes in treatment measures between baseline and 6 months will be compared between intervention and control groups to identify any clinically significant differences using ANCOVA regression with patients results at baseline fitted as a covariate. The model residuals will be checked to look for outliers and bimodality. Leave-one-out sensitivity analyses will be used to look for patients that strongly influence the results.

To maintain a patient outcome focus, effect size will be estimated using Minimally Clinically Important Difference. Minimally Clinically Important Differences (MCID) of outcome variables are summarised in Table 4. The HADS MCID has been validated in a population of acute respiratory failure survivors, and showed comparable estimates across studies, follow-up and country [22]. The U.S. Department of Veterans Affairs advises that when using the PCL-5

**Table 4. MCID for secondary outcomes.**

| PICS Feature | Measure | MCID |
|---|---|---|
| Anxiety/Depression | HADS | $\geq$ 2.5 points(22) |
| PTSD | PCL-5 | $\geq$ 10 points(23) |
| Physical Function | EQ-5D-5L | $\geq$ 0.5 points(24) |

to monitor progress of PTSD during treatment, a change of 5–10 points is reliable, while a change of >10 points is considered clinically meaningful [23]. A review of MCID for EQ-5D found heterogeneity in the literature, with thresholds between 0.03–0.52 being reported [24]. A threshold of 0.5 was selected for this study to ensure only meaningful change would be reported.

### 4.3 Sample characteristics

The demographics and results of the Baseline Outcome Measures Bundle will be described using proportions and associated 95% confidence intervals and visually compared between intervention and control groups to check the integrity of the randomisation.

The overall characteristics of the sample will be described using summary statistics. This will help inform the generalisability of our results. The number of patients approached and consented will be tabulated, together with the number who dropped out or died. If 40% of more of patients drop-out, we will use a logistic regression model to identify potential predictors of drop-out using the variables collected at baseline. The amount of wave and item-missing data will be tabulated by treatment group. In a sensitivity analysis, we will impute item missing data using multiple imputation.

There are no planned subgroup analyses.

Data will be exported to R (version 4.2.0 or higher) for analysis. Initial results will be created using a scrambled treatment group with the real analysis being created once the investigators agree that the proposed statistical analysis, tables, and graphs are complete and appropriate [25].

### 4.4 Data management and security

All data will be treated in confidence and only made accessible to members of the research team on an as needs basis. All paper records will be stored in a locked filing cabinet, and any electronic databases will be stored on password protected computers. For dissemination of results, participants will only be referred to in the coded form e.g. "Participant A", "Participant B" etc. All records will be destroyed (permanently deleted or shredded as appropriate) after 15 years, as per 'Good Clinical Practice' (GCP) guidelines.

Electronically collected data will be directly entered into REDCap, hosted on a Queensland Health server. Hard copy responses will be manually entered into REDCap by a member of the research team. Once recruitment is complete, Data will be cleaned and uploaded to statistical software package R for analysis.

Findings will be published in an academic journal and presented in scientific meetings. Deidentified data will be made publicly available for verification via the Open Science Framework.

### 5. Discussion

While the healthcare burden associated with PICS is clear, and an improved way to prevent, detect and treat the syndrome is needed, current literature is yet to identify the best approach to post-ICU follow-up and care. Most studies advocate for some form of multidisciplinary approach which is sensitive to the complex nature of the patient population, but more evidence is needed to support clinicians in identifying which patients to follow-up, in what setting, and what services to provide.

EPARIS will explore whether an early psychiatric review is an acceptable contribution to follow-up, and if the findings are suggestive of a possible clinical benefit. It will inform the design of larger clinical studies seeking to confirm whether the intervention improves patient outcomes.

The clinic involved in this study is one of few post-ICU clinics currently operating in Australia. However, there has been growing interest in post-ICU clinics in Australia, with new clinics being planned. As such, the findings of this study provide an opportunity to inform post-ICU care both locally and nationally. The strengths of EPARIS are the prospective, longitudinal design with a control population and its use of validated post-ICU outcome measures.

## 6. Limitations

As a pilot study with a small sample, this study is not powered to be able to identify clinical benefit, and while the participating clinic does accept referrals from other ICUs, the findings may not be generalisable to other settings given the heterogeneous nature of ICU populations. These limitations could be addressed by a subsequent multisite study adequately powered to evaluate the efficacy of the treatment on improving patient outcomes. The design and sample size for such a study would be greatly informed by the findings of this pilot study.

Some aspects of the feasibility and acceptability of the intervention would be better explored in a subsequent or nested qualitative study to compliment EPARIS, but a quantitative approach has been selected for this pilot study to estimate effect size and assist the research team in planning subsequent larger studies. Further, feasibility and acceptability to other stakeholders such as family members and general practitioners should be explored in subsequent studies.

## 7. Implications for practice

Improving post-ICU outcomes for patients with features of PICS has proven challenging. A multidisciplinary approach to post-ICU follow-up is commonly advocated for, with many health services providing this through a nurse-led post-ICU clinic. With evidence for post-ICU interventions currently being mixed and limited, the best approach remains unclear. This study will provide the first evidence as to whether there is a role for an early psychiatric review in the post-ICU follow-up process and provide a framework for future studies that wish to replicate this intervention in larger, multisite studies. This will help ICUs in planning and implementing post-ICU care pathways and referral processes.

## 8. Conclusion

This protocol provides a pragmatic evaluation of the acceptability of introducing early psychiatric assessment into an existing post-ICU follow-up process, and if considered acceptable will inform future research into the efficacy and generalisability of the intervention.

## 9. Impacts

This protocol provides a framework for a practical psychiatric intervention that can be applied to a post-ICU population and will evaluate whether that intervention will be acceptable to patients. The findings will inform larger multisite studies in demonstrating whether the intervention is effective and generalisable.

## Supporting information

**S1 Checklist. Spirit guidelines checklist.**
(DOCX)

**S1 Appendix. Initial survey documents.**
(DOC)

**S2 Appendix. Follow-up survey documents.**
(DOC)

**S1 File.**
(DOCX)

## Author Contributions

**Conceptualization:** Dylan Flaws, Chelsea Allen, Stuart Baker, Adrian Barnett, Olivia Metcalf, Hamish Pollock, Mahesh Ramanan, Alexis Tabah, Tracey Varker.

**Formal analysis:** Adrian Barnett.

**Funding acquisition:** Dylan Flaws.

**Methodology:** Dylan Flaws, Chelsea Allen, Stuart Baker, Adrian Barnett, Olivia Metcalf, Hamish Pollock, Mahesh Ramanan, Alexis Tabah, Tracey Varker.

**Project administration:** Dylan Flaws, Chelsea Allen.

**Supervision:** Dylan Flaws, Stuart Baker, Adrian Barnett, Olivia Metcalf, Mahesh Ramanan, Alexis Tabah, Tracey Varker.

**Writing – original draft:** Dylan Flaws.

**Writing – review & editing:** Dylan Flaws, Chelsea Allen, Stuart Baker, Adrian Barnett, Olivia Metcalf, Hamish Pollock, Mahesh Ramanan, Alexis Tabah, Tracey Varker.

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
