## [Decision Letter · Decision Letter 0]

6 Jan 2023

PONE-D-22-25386A Protocol for a Pilot Randomised Controlled Trial of an Early Psychiatric Assessment, Referral, and Intervention Study (EPARIS) for Intensive Care PatientsPLOS ONE

Dear Dr. Dylan Finlay Flaws,

Thank you for submitting your manuscript to PLOS ONE. After careful consideration, we feel that it has merit but does not fully meet PLOS ONE’s publication criteria as it currently stands. Therefore, we invite you to submit a revised version of the manuscript that addresses the points raised during the review process.

We look forward to receiving your revised manuscript.

Kind regards,

Alison Wang

Academic Editor

PLOS ONE

Journal Requirements:

3. We note that the original protocol that you have uploaded as a Supporting Information file contains an institutional logo. As this logo is likely copyrighted, we ask that you please remove it from this file and upload an updated version upon resubmission.

Additional Editor Comments:

1. Please further detail the methods of this study, including the methods of randomization, concealment, blinding.

2. please add one section to elaborate how to maintain the intervention fidelity of this study.

Reviewers' comments:

Reviewer's Responses to Questions

**Comments to the Author**

1. Does the manuscript provide a valid rationale for the proposed study, with clearly identified and justified research questions?

Reviewer #1: Yes

Reviewer #2: Yes

2. Is the protocol technically sound and planned in a manner that will lead to a meaningful outcome and allow testing the stated hypotheses?

Reviewer #1: Yes

Reviewer #2: Yes

3. Is the methodology feasible and described in sufficient detail to allow the work to be replicable?

Reviewer #1: Yes

Reviewer #2: Yes

4. Have the authors described where all data underlying the findings will be made available when the study is complete?

Reviewer #1: No

Reviewer #2: No

5. Is the manuscript presented in an intelligible fashion and written in standard English?

Reviewer #1: Yes

Reviewer #2: Yes

6. Review Comments to the Author

You may also provide optional suggestions and comments to authors that they might find helpful in planning their study.

Reviewer #1: This is a protocol paper for a pilot trial. Keeping up with the goal of the pilot trial the paper describes what the feasibility and acceptability of the pilot RCT type paper should target. My suggestion to only change is that author must either remove the ANCOVA analysis description or must state what is the purpose of that analysis. As a pilot RCT this must not target for establishing or testing any statistical anything.

The other revision I suggest is the writing style. For example, almost all sections consist of too many small paragraphs each consisting of 2-4 sentences. Also, provide section numbers for ease of readability and references.

Reviewer #2: This paper will contribute to a pragmatic evaluation of the acceptability of introducing early psychiatric assessment into an existing post-ICU follow-up process, and if considered acceptable, will inform future research into the efficacy and generalisability of the intervention. However, some changes are advised in order to improve the transparency for the reader:

1. -Was the sample size calculated upon previous studies? It remains unclear whether appropriate power calculations were made prior to recruiting patients. Please take this up in your methods section or in the discussion.

2. can you provide further details regarding how to recruit participants and research team members?

3. Page 10: No cognitive impairment that prevents participation: please be concise, what scale are you talking about here? It would be good to name this scale and add the appropriate reference of this scale.

4. What will be the primary outcomes and the secondary outcomes for this pilot RCT, and how to measure these outcomes? Please provide more details about the outcome measurements.

5. Please provide information about the roles of research team members. For example, who will be responsible for the intervention delivery? Who will be responsible for the data collection?

6. as stated in Page 17:” Estimate the effect size of any treatment effects to inform power calculations of future studies evaluating whether the intervention is beneficial”, how to calculate the effect size of any treatment effects? Are you going to use cohen’s d or Hedges's g?

7. as stated in Page 17 “The amount of wave and item-missing data will be tabulated by treatment group. We will impute item missing data using multiple imputation.” Please justify why multiple imputation will be used to deal with the missing data, instead of another commonly used method- last observation carried forward (LOCF).

8. please provide the information of “Data management”. Who can access all the collected data? And where are you going to store and dispose of the research data hard copies?

---

## [Author Response · Author response to Decision Letter 0]

23 Jan 2023

The authors would like to thank the reviewers for their comments and feedback. We have reviewed the reviewers' comments and suggestions and made a series of changes to the manuscript in light of these which we feel strengthen and clarify the paper. In addition, the authors have decided to remove the surveys from supporting documents due to concerns publishing these may contradict the licensing for some of the underlying clinical tools being used. All tools used in the surveys are still described and cited. 

Thank you again for taking the time to review this manuscript, and we hope that the changes summarized below have addressed your comments.

Warm Regards,

A/Prof. Dylan Flaws

Reviewer #1: This is a protocol paper for a pilot trial. Keeping up with the goal of the pilot trial the paper describes what the feasibility and acceptability of the pilot RCT type paper should target. My suggestion to only change is that author must either remove the ANCOVA analysis description or must state what is the purpose of that analysis. As a pilot RCT this must not target for establishing or testing any statistical anything.

We acknowledge the reviewer’s concerns, as while Statistical analysis can be used in pilot studies and be appropriate, problems can arise when people create p-values and start testing hypotheses. We have clarified both in the abstract, and in section 4.1 – Outcome Measures that we won’t test hypotheses in this study, but we still want to fit appropriate statistical models to get an idea of the size of the effect and best inform the effect size for the larger sample size calculation.

The other revision I suggest is the writing style. For example, almost all sections consist of too many small paragraphs each consisting of 2-4 sentences. Also, provide section numbers for ease of readability and references.

The manuscript has been revised to improve readability with better paragraphing and section numbers.

Reviewer #2: This paper will contribute to a pragmatic evaluation of the acceptability of introducing early psychiatric assessment into an existing post-ICU follow-up process, and if considered acceptable, will inform future research into the efficacy and generalisability of the intervention. However, some changes are advised in order to improve the transparency for the reader:

1. -Was the sample size calculated upon previous studies? It remains unclear whether appropriate power calculations were made prior to recruiting patients. Please take this up in your methods section or in the discussion.

This is a convenience sample based on the availability of the psychiatrist to administer the intervention for the purposes of this study for 12 months. This has now been clarified 

in Section 3.4 – Participants and Sample Size

2. can you provide further details regarding how to recruit participants and research team members?

This is now described in more detail in section 3.4 – Study Team, Participants and Sample Size

3. Page 10: No cognitive impairment that prevents participation: please be concise, what scale are you talking about here? It would be good to name this scale and add the appropriate reference of this scale.

This has been reworded as “and able to provide informed consent”

4. What will be the primary outcomes and the secondary outcomes for this pilot RCT, and how to measure these outcomes? Please provide more details about the outcome measurements.

The authors have intentionally avoided describing a “primary outcome measure” due to this being a pilot RCT. However, section 4.1 “Outcome Measure” has been reworded to try and clarify which outcome measures are being collected, and with which measurement tools. For further clarity, these have now been subdivided into “4.1.1 - Feasibility and acceptability outcomes”, and “4.1.2 – Treatment outcomes”.

5. Please provide information about the roles of research team members. For example, who will be responsible for the intervention delivery? Who will be responsible for the data collection?

This is now better described in section 3.4 – Study Team, Participants and Sample Size.

6. as stated in Page 17:” Estimate the effect size of any treatment effects to inform power calculations of future studies evaluating whether the intervention is beneficial”, how to calculate the effect size of any treatment effects? Are you going to use cohen’s d or Hedges's g?

The purpose of this pilot is to inform subsequent studies that will inform patient care. As such, our intention is to keep all measures focused on patient outcome. For this reason, the investagors have elected for MCID rather than Cohen’s d or Hedge’s g as a measure of effect size. Section 4.1.2 clarifies the rationale for this choice and refers the reader to Table 4 for the MCID of each outcome variable.

7. as stated in Page 17 “The amount of wave and item-missing data will be tabulated by treatment group. We will impute item missing data using multiple imputation.” Please justify why multiple imputation will be used to deal with the missing data, instead of another commonly used method- last observation carried forward (LOCF).

Section 4.3 The authors elected for multiple imputation as LOCF relies on a strong assumption of no change and makes no use of potentially valuable information about the participants and/or time. Simulation studies have shown that it does not perform as well as multiple imputation for dealing with missing data (see for example DOI: 10.1002/sim.2099 and DOI: 10.1080/10543400903242795 and https://pubmed.ncbi.nlm.nih.gov/15889392/).

8. please provide the information of “Data management”. Who can access all the collected data? And where are you going to store and dispose of the research data hard copies?

Section 4.4 – Data Management and Security now describes how physical and electronic data will be managed during the study, and how findings and deidentified data will be made available after study completion.

---

## [Decision Letter · Decision Letter 1]

23 Mar 2023

PONE-D-22-25386R1A Protocol for a Pilot Randomised Controlled Trial of an Early Psychiatric Assessment, Referral, and Intervention Study (EPARIS) for Intensive Care PatientsPLOS ONE

Dear Dr. Flaws,

Thank you for submitting your revised manuscript to PLOS ONE. After careful consideration, we feel that it has merit but does not fully meet PLOS ONE’s publication criteria as it currently stands. Therefore, we invite you to submit a new revised version to address the points raised during this round of reviews. Due to unavailability of previous reviewers, your manuscript was sent to two new reviewers who expressed a favourable opinion, but require further minor amendments and specifications before publication.

We look forward to receiving your revised manuscript.

Kind regards,

Armando D'Agostino

Academic Editor

PLOS ONE

Journal Requirements:

Reviewers' comments:

Reviewer's Responses to Questions

**Comments to the Author**

1. Does the manuscript provide a valid rationale for the proposed study, with clearly identified and justified research questions?

Reviewer #3: Yes

Reviewer #4: Yes

2. Is the protocol technically sound and planned in a manner that will lead to a meaningful outcome and allow testing the stated hypotheses?

Reviewer #3: Yes

Reviewer #4: Yes

3. Is the methodology feasible and described in sufficient detail to allow the work to be replicable?

Reviewer #3: Yes

Reviewer #4: Yes

4. Have the authors described where all data underlying the findings will be made available when the study is complete?

Reviewer #3: Yes

Reviewer #4: Yes

5. Is the manuscript presented in an intelligible fashion and written in standard English?

Reviewer #3: Yes

Reviewer #4: Yes

6. Review Comments to the Author

You may also provide optional suggestions and comments to authors that they might find helpful in planning their study.

Reviewer #3: I am late to this party. I wonder if the chosen acceptability outcome will really reflect acceptability. The analysis will, of course, be carried on people who accepted to participate in the study. Among those who refused, some may have issues of acceptance of a psychiatric intervention. I guess the authors will have the opportunity to discuss this in their final paper. A qualitative approach might provide a deeper understanding of the acceptability and the overall experience of meeting a psychiatrist.

Reviewer #4: Thank you for giving me the opportunity to review this protocol manuscript. The authors propose an interesting intervention that if feasible, could greatly impact the well-being of ICU patients.

I was however surprised that the authors went for a quantitative approach for this pilot study. With such a small expected sample size and given the objectives, focus groups or interviews with patients and healthcare providers might have been more pertinent. At the very least, I would recommend to carefully document any comments from patients, psychiatrist, and other healthcare provider to strengthen the evaluation of the feasibility and acceptability of the psychiatric intervention.

This being said, I appreciate how the authors remain humble and transparent regarding sample size, statistical power, and analyses. For instance, the sample size and participation rate estimation are very well described: “The feasible recruitment window is 12 months, with an estimated maximum possible sample size of 80. Assuming a 50% recruitment percentage, it is estimated that data will be collected from a sample of approximately 30 to 40 patients attending the Redcliffe ICU follow-up clinic over 12 months, with 15 to 20 allocated to the control and intervention arms respectively.” However, they do not mention any expectation regarding dropout between appointment and 6-months post-appointment time. I would advise to give an approximation of the expected dropout when describing the expected sample size.

I would advise to also measure the LEC-5 at 6-months post appointment to ensure that no new adverse life event could explain a big change at follow-up.

The authors mention how and when the psychiatrist will refer to the patients’ GP, and all in all it seems that a lot is put on the hands of the GPs. Thus, the authors might want to also include the GPs’ perspective when evaluating feasibility and acceptability of the intervention. Also, the “Health Service Use” variable results at follow-up will depend quite substantially on the reactivity of the patient’s GP.

Could the authors please provide a reference for the MCID they plan to use? And ideally mention whether and how those thresholds have been validated.

Other minor comments:

- I would advise the authors to avoid abbreviations in the Abstract or clearly state their meaning the first time they are mentioned in the abstract.

- Please provide the full name of the EQ5D-5L somewhere.

- In Table 3, the term “Systems” is not clear.

- “Participants will also give permission for research staff to access data from their clinical records and will complete a series of questionnaires as described below.” I think the authors mean “above”

7. PLOS authors have the option to publish the peer review history of their article (what does this mean?). If published, this will include your full peer review and any attached files.

Reviewer #3: **Yes: **Michael Saraga

Reviewer #4: **Yes: **Valerie Carrard

---

## [Author Response · Author response to Decision Letter 1]

5 Apr 2023

Thank you for the opportunity to revise this manuscript, and for the feedback from the reviewers. 

We acknowledge the limitations noted by both reviewers of a quantitative approach evaluating the acceptability of the intervention, and this will be reflected in the way the findings are interpreted and reported. We have made a small amendment to the acceptability question to provide the participant to specifically comment on whether they would recommend adding a psychiatric review to current clinic cares. The free text question at the bottom of the post-appointment questionnaire may provide some insights, but the authors feel a dedicated qualitative study is likely to be required following this study.

Similarly, the need to look to other important stakeholders such as GPs is also a very important point, and one the authors have discussed thoroughly. We looked at how we could modify the current study to capture this information but are concerned that it would not produce data of a sufficient quality to answer the question. As such, propose that a subsequent dedicated study will be required to explore the feasibility and acceptability of the intervention from the GP perspective, perhaps combined with the aforementioned qualitative study. We now discuss this in the limitations section.

We have attempted to address the other points raised by the reviewers, including clarifying the expected dropout rates and adding LEC-5 to the follow-up survey.

Thank you again for taking the time to review this protocol paper and constructive feedback.

---

## [Decision Letter · Decision Letter 2]

30 May 2023

PONE-D-22-25386R2A Protocol for a Pilot Randomised Controlled Trial of an Early Psychiatric Assessment, Referral, and Intervention Study (EPARIS) for Intensive Care PatientsPLOS ONE

Dear Dr. Flaws,

Thank you for submitting your revised manuscript to PLOS ONE. We invite you to submit a new version of the manuscript that addresses the two minor points raised during this round of the review process.

We look forward to receiving your revised manuscript.

Kind regards,

Armando D'Agostino

Academic Editor

PLOS ONE

Journal Requirements:

Reviewers' comments:

Reviewer's Responses to Questions

**Comments to the Author**

1. Does the manuscript provide a valid rationale for the proposed study, with clearly identified and justified research questions?

Reviewer #4: Yes

2. Is the protocol technically sound and planned in a manner that will lead to a meaningful outcome and allow testing the stated hypotheses?

Reviewer #4: Yes

3. Is the methodology feasible and described in sufficient detail to allow the work to be replicable?

Reviewer #4: Yes

4. Have the authors described where all data underlying the findings will be made available when the study is complete?

Reviewer #4: Yes

5. Is the manuscript presented in an intelligible fashion and written in standard English?

Reviewer #4: Yes

6. Review Comments to the Author

You may also provide optional suggestions and comments to authors that they might find helpful in planning their study.

Reviewer #4: I thank the authors for their answer to my comments and the revisions undertaken. I found that they addressed satisfactorily my concerns, except for two minor points:

1. I now see the references for the MCIDs and I would still encourage the author to mention their validation in the text. For instance, by stating that the HADS’ MCID was validated in a survivors of acute respiratory failure sample using distribution-based methods and that it showed comparable estimates across studies, follow-up, and country.

2. Mentioning a potential qualitative follow-up investigation is a good think. I would additionally indicate why a quantitative exploration is an important first step.

7. PLOS authors have the option to publish the peer review history of their article (what does this mean?). If published, this will include your full peer review and any attached files.

Reviewer #4: **Yes: **Valerie Carrard

---

## [Author Response · Author response to Decision Letter 2]

1 Jun 2023

To whom it may concern,

We hope the attached manuscript has addressed the comments and provided a more detailed justification for the MCIDs used in this study, as well as the rationale for a quantitative pilot study prior to a subsequent qualitative or nested study.

Reviewer #4: I thank the authors for their answer to my comments and the revisions undertaken. I found that they addressed satisfactorily my concerns, except for two minor points:

1. I now see the references for the MCIDs and I would still encourage the author to mention their validation in the text. For instance, by stating that the HADS’ MCID was validated in a survivors of acute respiratory failure sample using distribution-based methods and that it showed comparable estimates across studies, follow-up, and country.

Thank you, we have now elaborated in section 4.1.2 how the MCID values listed in table 4 were validated.

2. Mentioning a potential qualitative follow-up investigation is a good think. I would additionally indicate why a quantitative exploration is an important first step.

We have elaborated in section 6 that a quantitative pilot has been selected in the first instance to inform subsequent larger studies, with subsequent or nested qualitative studies intended.

Thank you for taking the time to review this manuscript, and please let us know if you have any further questions.

Warm Regards,

A/Prof. Dylan Flaws

---

## [Editor Report · Decision Letter 3]

6 Jun 2023

A Protocol for a Pilot Randomised Controlled Trial of an Early Psychiatric Assessment, Referral, and Intervention Study (EPARIS) for Intensive Care Patients

PONE-D-22-25386R3

Dear Dr. Flaws,

We’re pleased to inform you that your manuscript has been judged scientifically suitable for publication and will be formally accepted for publication once it meets all outstanding technical requirements.

Kind regards,

Armando D'Agostino

Academic Editor

PLOS ONE
---

## [Editor Report · Acceptance letter]

19 Jun 2023

PONE-D-22-25386R3 

A Protocol for a Pilot Randomised Controlled Trial of an Early Psychiatric Assessment, Referral, and Intervention Study (EPARIS) for Intensive Care Patients 

Dear Dr. Flaws:

I'm pleased to inform you that your manuscript has been deemed suitable for publication in PLOS ONE. Congratulations! Your manuscript is now with our production department. 

Kind regards, 

on behalf of

Dr. Armando D'Agostino 

Academic Editor

PLOS ONE